# *Caveat emptor:* On the Need for Baseline Quality Standards in Computer Vision Wood Identification

Prabu Ravindran [1,2,*] and Alex C. Wiedenhoeft [1,2,3,4,5]

1 Department of Botany, University of Wisconsin, Madison, WI 53706, USA; alex.c.wiedenhoeft@usda.gov
2 Center for Wood Anatomy Research, USDA Forest Service, Forest Products Laboratory, Madison, WI 53726, USA
3 Department of Forestry and Natural Resources, Purdue University, West Lafayette, IN 47907, USA
4 Department of Sustainable Bioproducts, Mississippi State University, Starkville, MS 39762, USA
5 Departamento de Ciências Biológicas (Botânica), Universidade Estadual Paulista, Botucatu, Botucatu 18610-307, SP, Brazil
* Correspondence: pravindran@wisc.edu

**Abstract:** Computer vision wood identification (CVWID) has focused on laboratory studies reporting consistently high model accuracies with greatly varying input data quality, data hygiene, and wood identification expertise. Employing examples from published literature, we demonstrate that the highly optimistic model performance in prior works may be attributed to evaluating the wrong functionality—wood specimen identification rather than the desired wood species or genus identification—using limited datasets with data hygiene practices that violate the requirement of clear separation between training and evaluation data. Given the lack of a rigorous framework for a valid methodology and its objective evaluation, we present a set of minimal baseline quality standards for performing and reporting CVWID research and development that can enable valid, objective, and fair evaluation of current and future developments in this rapidly developing field. To elucidate the quality standards, we present a critical revisitation of a prior CVWID study of North American ring-porous woods and an exemplar study incorporating best practices on a new dataset covering the same set of woods. The proposed baseline quality standards can help translate models with high in silico performance to field-operational CVWID systems and allow stakeholders in research, industry, and government to make informed, evidence-based modality-agnostic decisions.

**Keywords:** wood identification; computer vision; machine learning; XyloTron; best practices

## 1. Introduction

The paucity of wood identification expertise [1,2] has spurred interest in automated wood identification technologies, especially in the context of combating illegal logging. Among the technologies considered [3], computer vision-based wood identification (CVWID) has been widely studied [4–13] and is highly effective [14], field-deployable [9,15], and in rare cases, field-tested [16]. Additionally, the democratization of CVWID technologies through affordable, open-source, do-it-yourself, hardware (e.g., the XyloTron (XT) platform [17]; the XyloPhone [18]) and robust, efficient software implementations of standard computer vision (CV) and machine learning (ML) techniques (e.g., [19–22]) can enable robust, multi-point monitoring of the wood and wood products value chain. Despite high in silico accuracies reported in virtually all prior studies (ref. [23] provides a literature survey), field-operational CVWID systems with acceptable real-world performance have remained elusive or unreported. To inform and enable an evidence-based transition of CVWID systems from the laboratory to the real world, a critical revisitation of the state-of-the-practice is required to identify assumptions, blind spots, and bottlenecks, and to explicitly articulate the foundational principles and best practices for CVWID systems.

### 1.1. The Central Dogma of ML

The objective of ML is the development of models with the desired functionality on a training dataset that perform acceptably on testing data that were not used for training (i.e., avoiding overfitting to the training data and performing acceptably on the test dataset [24]). Careful collection of high-quality datasets (with a sufficient number of samples at the correct atomicity) that capture the classification-relevant features is needed for effective training and valid evaluation of ML models with the desired domain-specific functionality, and in many cases, the availability of high-quality data may be more important than sophisticated algorithms [25]. Given a dataset that captures the expected/typical inter-and intra-class variations, mutual exclusivity between the training and testing datasets at the correct sample atomicity is a foundational requirement for training and evaluating valid ML models.

### 1.2. Core Tenets of CVWID

The functionality required from a CVWID system is image-based wood identification—given a wood specimen (e.g., wood block, log, tree stump), classify it into one among a known set of classes using the wood anatomy visible in one or more images from the specimen. The features (hand-crafted or learned) used for classification must correlate well with the relevant inter-class wood anatomy differences that allow accurate, generalizable classification while being able to handle expected intra-class variations of the features. Given these requirements, two foundational CVWID principles become immediately apparent:

1.  Sample atomicity at the level of individual trees is mandatory for a valid training and evaluation methodology. Violation of this requirement leads to systems that are evaluated for the wrong functionality—individual tree recognition—instead of the desired wood species or genus identification functionality (see Figure 1 for example scenarios), and
2.  Consistent sample preparation and imaging that enables the capture of images that show the relevant discriminative wood anatomy, thereby enabling models to learn and use robust features based on the arrangement, abundance, and structural patterns of wood anatomical features such as vessels, rays, and axial parenchyma, and reduces the chances of learning spurious or non-robust correlations (e.g., color, systematic defects in surface preparation).

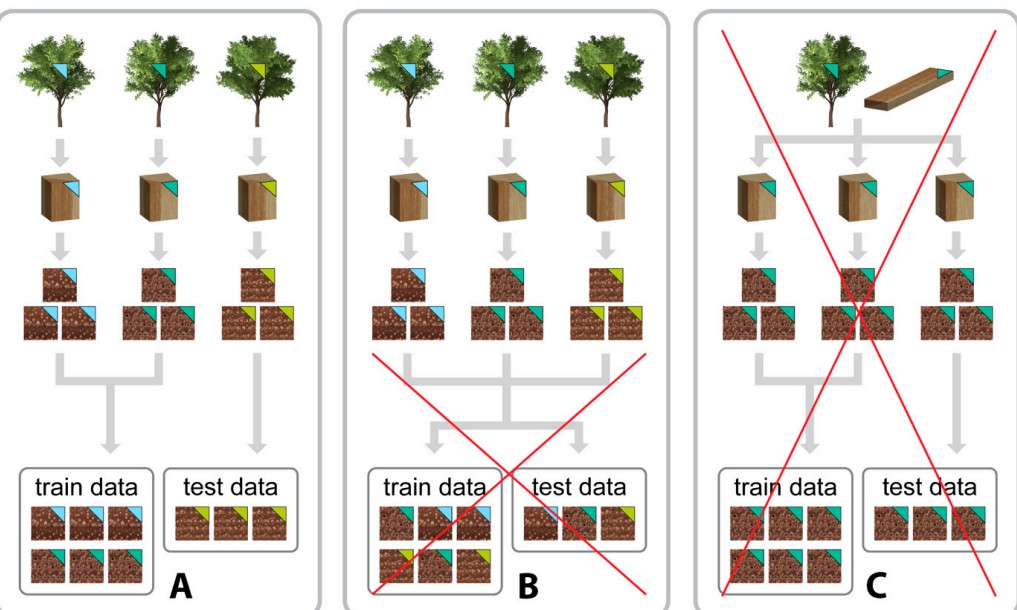

**Figure 1.** Illustration of three data hygiene scenarios for computer vision-based wood identification (CVWID) training and evaluation using a single class with three specimens. (**A**) Dataset development

in accordance with the foundational principles, wherein multiple unique specimens (from distinct trees) are used for dataset collection and mutual exclusivity at the specimen level between the training and testing folds is maintained. (**B**) A scenario where the evaluation is flawed because of specimen level data leakage between the training and testing folds even though multiple unique specimens (from different trees) were used in the dataset. Correct partitioning into training and testing folds (as shown in (**A**)) salvages this scenario. (**C**) A scenario that violates the baseline quality standards for dataset collection precluding valid performance evaluation—this is not salvageable and typically occurs when specimens from teaching collections or other non-scientific collections (in which individual specimens are not traceable) are used for data collection.

### 1.3. The CVWID Lifecycle

The design, implementation, and evaluation lifecycle of a CVWID system comprise the prototyping and iterative refinement of the following stages (Figure 2):

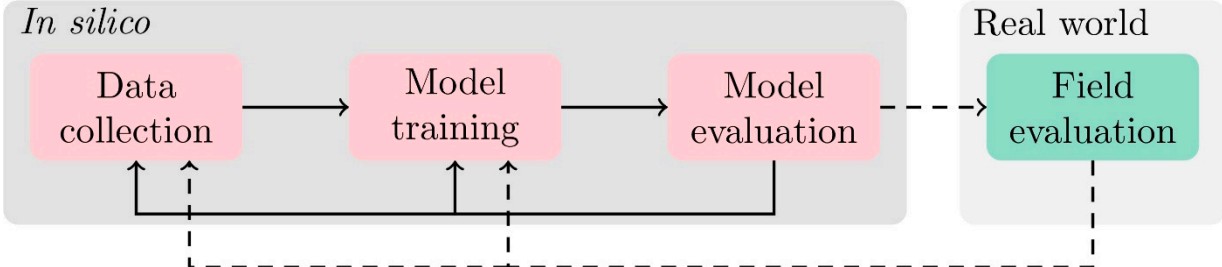

**Figure 2.** Schematic of the CVWID lifecycle. Most prior works have been restricted to in silico model development and evaluation based on a limited number of unique xylarium specimens that do not allow valid specimen level mutual exclusivity between the training and testing data. The sufficiency of CVWID systems can be validated only with context-informed field evaluation but this is mostly unexplored in prior works (see text for exceptions). The dashed lines are the paths not explored in existing works.

#### 1.3.1. Context Definition

The scope of a field operational CVWID system is defined by the set of woods the system must identify (e.g., continental, national, regional scale); the granularity of the identification (e.g., species/genus level identification (multi-class classification); endangered/controlled vs. non-controlled (binary classification)); and, the risk of a misidentification (not all misidentifications are considered equal, e.g., misidentifications within a genus versus out-of-genus misidentification; misidentification of a high-value timber as low value versus the converse). Expertise in wood anatomy and compliance enforcement is critical for identifying and defining a practical, well-posed context.

#### 1.3.2. Data Collection

Data collection for CVWID comprises the following steps: specimen acquisition, specimen surface preparation, and prepared specimen surface imaging. To enable the training of accurate CVWID models and their valid evaluation, the images in the dataset must be obtained from verified specimens, consistently capture the relevant wood anatomy, and the dataset as a whole must capture the typical inter- and intra-class variability.

#### 1.3.3. Model Training

The two components of a CVWID model are a feature extractor (which computes a "signature" that encodes the wood anatomy patterns visible in an image) and a classifier (a learnable, parametrized function that maps image signatures to class labels). Models used for identification can be shallow (hand-crafted feature extractors with learned classifiers) or deep (automated end-to-end parameter learning of both the feature extractor and the classifier [26]), with deep learning models [27] gaining traction due to their ability to

automatically learn powerful data-driven, task-relevant features. Model training is the process of estimating the parameters of the feature extractor and the classifier by optimizing a goodness-of-fit (e.g., categorical cross-entropy loss [28]) criterion on a training dataset.

### 1.3.4. Model Evaluation

Analysis of model generalization to unseen data (data not used for model training) is at the core of ML, and for scientific validity and practical relevance it must be performed on data that are mutually exclusive from the training data at the individual tree level. Cross-validation, with splits made at the correct sample atomicity, is a common technique that can be used for in silico evaluation of model generalizability. Model training and evaluation are critically dependent on dataset quality and as with any computational scheme the potential impact of garbage in, garbage out [29] should be considered.

### 1.3.5. Field Evaluation

The gold standard that establishes the sufficiency of any wood identification system is real-world field evaluation (i.e., deployed and evaluated on multiple specimens at ports or points in the supply chain, especially with multiple hardware instantiations). A wood identification system should only be deemed operational if it has acceptable context-specific performance metrics (e.g., accuracy, misclassification cost) when deployed in the field. The goal of research endeavors in CVWID should be the realization of field-operational systems, not merely the production of models with high in silico overall accuracy metrics that may or may not generalize to new specimens; different system users; new hardware instantiations; and/or, reasonable deviations in specimen preparation.

### 1.4. Challenges and State-of-the-Practice in CVWID

### 1.4.1. Sourcing, Scarcity, and Data Quality of CVWID Data

Access to (a reasonable number of) scientifically/forensically validated specimens is a fundamental challenge for CVWID development. Two "active" methods for sourcing specimens are targeted field expeditions and the acquisition of commercial specimens, with both approaches being resource-intensive, logistically challenging, and highly dependent on botanical expertise. To the best of our knowledge, most prior works have leveraged xylarium specimens for the development and evaluation of CVWID models; studies using non-xylarium specimens suffer from an additional level of potential unreliability, that is, the specimens from which images were collected are not scientifically validated, and/or traceable. While xylaria can be a convenient source for specimens, most xylaria accessions were collected for botanical and teaching purposes and often contain few specimens (from different individual trees) per species. The lack of sufficient numbers of mutually exclusive specimens precludes a valid training and evaluation methodology with mutual exclusivity at the individual tree level. For example, one cannot conduct five-fold cross-validation at the specimen level with only four specimens per class. Assuming the availability of a sufficient number of specimens, this raises the question of "how well do models trained on xylarium specimens generalize to unseen commercial specimens?" The criteria used to collect xylarium specimens differ from those used in harvest for commercial purposes, which prioritizes tree form, size, volume, and ease of harvest rather than the illustration of taxon-dependent typical wood anatomy. The impact of this potential distributional shift between training and testing data has not been characterized in prior CVWID works [30,31], though it was raised as a concern in [16]. With the hope of comprehensively capturing the neighborhood of training data points and closing the gap between the training and testing distributions, most prior works employ data augmentation [32], but the purpose or value of augmented data when specimens contribute images both to the training and testing datasets is unclear. In fact, most prior works that use only a handful of xylarium samples (which are typically small and may or may not capture the feature variability typical for the species) with data augmentation and do not enforce individual-tree-level mutual exclusivity for the training/testing splits, the high accuracies reported are not

surprising and should be expected as training and testing happen on (a small portion of) the same trees (i.e., a highly increased risk of overfitting). The sufficient number of specimens required to capture the typical variability within a species is unknown. Ref. [33] provides an assertion of 10 specimens and 100 images per specimen for rosewood, but these numbers are likely to be species-dependent as a function of wood anatomical variability. As an extreme example, when images from only one specimen are used to train and evaluate models, the implicit assumption is that the one specimen captures the necessary variability for the species.

### 1.4.2. Logistics of Field Testing

The execution of robust field evaluation requires independent forensic validation of the specimens being tested (e.g., by traditional light microscopic identification in the laboratory or by on-site identification by a qualified expert) in order to check the prediction of the trained model. Presumably, the logistical challenges of connecting field specimens to (scarce) wood identification expertise for forensic validation have been a bottleneck for conducting rigorous field evaluation of CVWID systems. With the exception of [16], who reported performance metrics for the field evaluation of 15 commercial Ghanaian woods, albeit on relatively few specimens, all prior works have been limited to in silico experiments (using cross-validation and variants thereof) with reports of high accuracies being the norm. Ref. [13] reported metrics for a surrogate field-testing scenario where the trained model was evaluated on specimens from a separate xylarium that did not contribute data to the training data. While this approach is a promising first step that considered the variability in a multi-operator, multi-site, multi-hardware instantiation deployment, mutual exclusivity between the surrogate field specimens and the training data was not explicitly investigated (mutual exclusivity between training and testing data could have been violated if a collector provided specimens, for the taxa considered, from the same tree to more than one of the participating xylaria). Another limitation of the surrogate field evaluation is that it does not address the issue of generalization of models trained on xylarium specimens to (new) field specimens—an issue of great significance for the realization of field-deployable CVWID systems given the revolution that is taking place in the digitization of herbarium specimens (e.g., [34–36]).

The democratization of high-quality implementations of CV/ML models and algorithms has considerably eased their adoption, but the practice of scientifically rigorous applied ML methodology is the responsibility of the practitioner, and evaluating such methods is the responsibility of any adopter of such technology. The challenges identified above establish the clear need to articulate a minimum set of criteria for building CVWID systems to enable objective evaluation of prior studies and benchmarking future developments so that the perils and pitfalls of garbage-in fiction-out (GIFO)—wherein spurious associations within a flawed or inadequate dataset that are otherwise not generalizable to the real world are learned by a model—hypothesized in [18] are avoided. In this paper, we critically analyze a study [37] on ten North American hardwoods and demonstrate that it incorporates each of the methodological concerns identified above. We contrast the methodology in [37] with a demonstration of well-performed dataset collection, model training, and model performance evaluation on a new dataset for the same ten classes of woods. The motivation for this work was not the development of a novel ML model or training recipe, but rather an explicit reiteration of the basic requirements for valid ML methodology in the context of CVWID, with clear examples of how failing to adhere to sound ML methodology yields results that, while demonstrating high in silico accuracies, lack appreciable real-world functionality or relevance. We see the critical need for objective comparison of claims in the rapidly developing field of CVWID as a clear case of *caveat emptor*, and to enable a "buyer" (e.g., researcher, practitioner, policy-maker, compliance officer) to make informed, evidence-based decisions, we propose a minimal set of necessary baseline quality standards to ensure the rigorous development, evaluation, and real-world

deployment of these systems that, by analogy, are directly applicable to modalities other than CVWID alone.

## 2. Materials and Methods

### 2.1. Datasets

Four macroscopic wood image datasets (LD-train, LD-test, XT-train, and XT-test) were employed to elucidate the core tenets of CVWID and to motivate the need for baseline quality standards. All four datasets comprise images of the transverse surfaces of wood specimens from North American ring-porous hardwoods that were grouped into 10 anatomically relevant classes.

### 2.1.1. LD-Train

Ref. [37] reported that the 10-class image dataset subtending their work was obtained from wood specimens whose transverse surfaces were exposed using razor blade cuts and imaged using an unspecified commercial lens affixed to a mobile phone. The origin and traceability of the wood specimens (other than from the teaching collection at Mississippi State University, Department of Sustainable Bioproducts), the number of unique wood specimens that were imaged, and the pixel resolution (microns/pixel) of the imaging system were not reported. This dataset, LD-train, comprised 1709 (of 1869 reported) RGB images each of 3024 × 3024 pixels (download URL: https://ir.library.msstate.edu/handle/11668/18480, accessed on 13 August 2020, see comment in Supplement).

### 2.1.2. LD-Test

The LD-test dataset was acquired by subjecting the transverse surfaces of wood specimens to progressive coarse to fine grit sanding to expose the wood anatomical features and imaging the prepared surfaces at 1.0× zoom with a 14× Olloclip array [18] attached to an iPhone XR mobile phone. Fifty scientifically verified specimens from the MADw and SJRw xylaria (nomenclature for xylaria per Index Xylariorum 4.1 (2021) [38]) were employed to produce the 250 RGB images (3024 × 3024 pixels each) in the LD-test dataset. Further details on making the image magnification and resolution for the LD-test dataset comparable to that for the LD-train images can be found in the Supplement.

### 2.1.3. XT-Train

The open-source XyloTron (XT) system [17] was used to image the prepared transverse surfaces of 219 specimens from the MADw and SJRw xylaria at the USDA Forest Products Laboratory to produce the XT-train dataset containing 2635 images. Each 2048 × 2048-pixel XT image shows 6.35 × 6.35 mm of tissue.

### 2.1.4. XT-Test

The XT-test dataset, containing 339 images, was obtained by imaging 71 specimens from the David A. Kribs wood collection (PACw) and the teaching collection at Mississippi State University (with light microscopy validation of the teaching specimens). The surfaces of the wood specimens used for the XT-train and XT-test datasets were subjected to progressive coarse to fine grit sanding to expose the wood anatomical features.

The identification of all specimens used in the XT-train, XT-test, and LD-test datasets was confirmed by a wood identification expert using a traditional wood anatomy-based approach prior to inclusion in a dataset. The LD-test and XT-test datasets are mutually exclusive from the LD-train and XT-train datasets respectively i.e., the specimens in the LD-test dataset did not contribute images to the LD-train dataset and specimens in the XT-test dataset did not contribute images to the XT-train dataset. This specimen level mutual exclusivity between the training and testing datasets is necessary for valid ML methodology. A summary of the image and specimen counts for the four datasets is provided in Table 1. The class-wise montages of the central patches for every image in the LD-train and XT-train datasets are presented in the Supplement.

**Table 1.** Class-wise image counts, including specimen counts (in parentheses) for the four datasets. The number of specimens used in LD-train was not reported in [37].

| Class Label | LD-Train | LD-Test | XT-Train | XT-Test |
|:---:|:---:|:---:|:---:|:---:|
| Celtis | 175 | 25 (5) | 300 (19) | 14 (3) |
| Fraxinus | 229 | 25 (5) | 300 (27) | 18 (4) |
| Gleditsia | 134 | 25 (5) | 121 (12) | 15 (3) |
| Maclura | 180 | 25 (5) | 203 (18) | 23 (5) |
| Morus | 250 | 25 (5) | 300 (23) | 25 (5) |
| QuercusR | 153 | 25 (5) | 300 (30) | 126 (27) |
| QuercusW | 183 | 25 (5) | 300 (33) | 40 (8) |
| Robinia | 135 | 25 (5) | 249 (17) | 15 (3) |
| Sassafras | 125 | 25 (5) | 262 (19) | 33 (7) |
| Ulmus | 145 | 25 (5) | 300 (21) | 30 (6) |
| Total | 1709 | 250 (50) | 2635 (219) | 339 (71) |

### 2.1.5. Species Composition

The North American hardwood taxa used in the four datasets were categorized into one of the following ten classes: eight genus level classes "Celtis", "Fraxinus", "Gleditsia", "Maclura", "Morus", "Robinia", "Sassafras" and "Ulmus"; and two sub-generic classes of the genus *Quercus*, namely "QuercusR" and "QuercusW" for red and white oak groups, respectively. Each class in LD-train was reported as comprised of exactly one species [37]. The species used in LD-test, XT-train, and XT-test were also grouped into the same set of classes such that congeneric species comprising a class were indistinguishable based on their macroscopic anatomical features. Class labels are reported without italics to distinguish when we discuss classes rather than botanical genera (as in [13]). Wood identification is typically accurate only at the genus level [39], except when macroscopic identification at the sub-generic level is possible, e.g., discriminating between the white and red oaks, and the grouping of species with macroscopic anatomical similarities is a practical, domain expertise-driven approach to capture the wood anatomical variability within a class and to build datasets with sufficient independent, unique specimens. The class labels and the species compositions for the four datasets are listed in Table S1 in the Supplement.

### 2.2. Modeling and Evaluation

#### 2.2.1. Evaluation of LD-Train

It is evident in Figure 1 of [37] that the specimen preparation was inadequate to produce images in which the wood anatomical features could be reliably observed. In order to understand their dataset prior to developing our own CVWID model with those data, we evaluated every image in LD-train for: whether it was the correct taxon; if at least approximately one-quarter of the image was knife prepared (vs. saw cut); whether at least approximately one-quarter of the image showed relevant wood anatomical detail; if the image showed no wood anatomical detail; whether the image was in focus; whether it was $3024 \times 3024$ pixels; and, whether an image was a duplicate of another image in the dataset.

#### 2.2.2. Model Architecture and Training

A convolutional neural network (CNN; [40]), with an ImageNet [41] pre-trained ResNet34 [42,43] backbone and a custom head (see [13,16] for architecture schematics), was trained separately on each of the LD-train and XT-train datasets using transfer learning [44]. The two-stage transfer learning methodology [45] comprised of freezing the pre-trained backbone weights while training the custom head weights followed by finetuning the weights of the entire network. The Adam optimizer [46] with simultaneous cosine annealing

of the learning rate and momentum [47] was employed for both stages. Random $2048 \times 768$ image patches were sampled from the training images, downsampled to $512 \times 192$-pixel images, and fed to the CNN in batches of size 16 with a data augmentation strategy that included horizontal/vertical flips, rotations (up to 5 degrees), and cutout [48]. When the image dimensions did not allow extraction of $2048 \times 768$-pixel patches zero-padding was employed (required for some LD-train images, but not needed for XT-train images) to generate patches with the required dimensions. The image resolution (microns per pixel) for LD-train was unspecified and hence it was not possible to stipulate training patch dimensions that covered tissue commensurate with the $2048 \times 768$-pixel patches extracted from the XT-train images. The model definition, training, and evaluation were performed using PyTorch [22] and scientific Python tools [20].

### 2.2.3. Performance Evaluation

The wood identification models were trained and evaluated in two ways:

#### Five-Fold Cross-Validation

A standard cross-validation analysis, where each fold serves as the testing set for a model trained on the remaining four-folds, was performed and the prediction accuracy and confusion matrix aggregated over the five folds are reported. The cross-validation models trained on LD-train and XT-train are referred to as LD5 and XT5 respectively. A valid cross-validation analysis requires specimen level mutual exclusivity between the folds. For LD-train, the absence of specimen level details i.e., the source specimen identifier for each image, precluded the stratification of cross-validation folds at the specimen level. The five-folds used for LD5 were obtained based only on class level stratification and unless every image in LD-train were from a different specimen one must infer a lack of the required mutual exclusivity at the specimen level. In contrast, the cross-validation folds for XT5 were obtained with the necessary specimen level mutual exclusivity. For XT5, the specimen level prediction was the majority of the predictions on (up to 5 randomly selected) images contributed by the specimen. Performance metrics (accuracy and confusion) at both the image and specimen level are reported for XT5 whereas only image-level metrics are reported (and possible) for LD5.

#### Surrogate Field Testing

Field models LDF and XTF were trained using all of LD-train and XT-train, respectively, and evaluated on LD-test and XT-test, respectively, as a surrogate for real-world field testing [13]. Specimen level accuracies and confusion matrices are reported for LDF and XTF as specimen level details are available for both LD-test and XT-test. Similar to XT5, the specimen level prediction for LDF and XTF was obtained as the majority of up to 5 images contributed by the specimen.

To summarize, LD-train is used to train LD5 and LDF, and LD-test is used to evaluate LDF. XT-train is used to train XT5 and XTF, and XT-test is used to evaluate XTF i.e., the field models are evaluated using specimens that did not contribute images to the training data. For reference, the schematic roadmap for the analyses performed with the four datasets is provided in Figure 3.

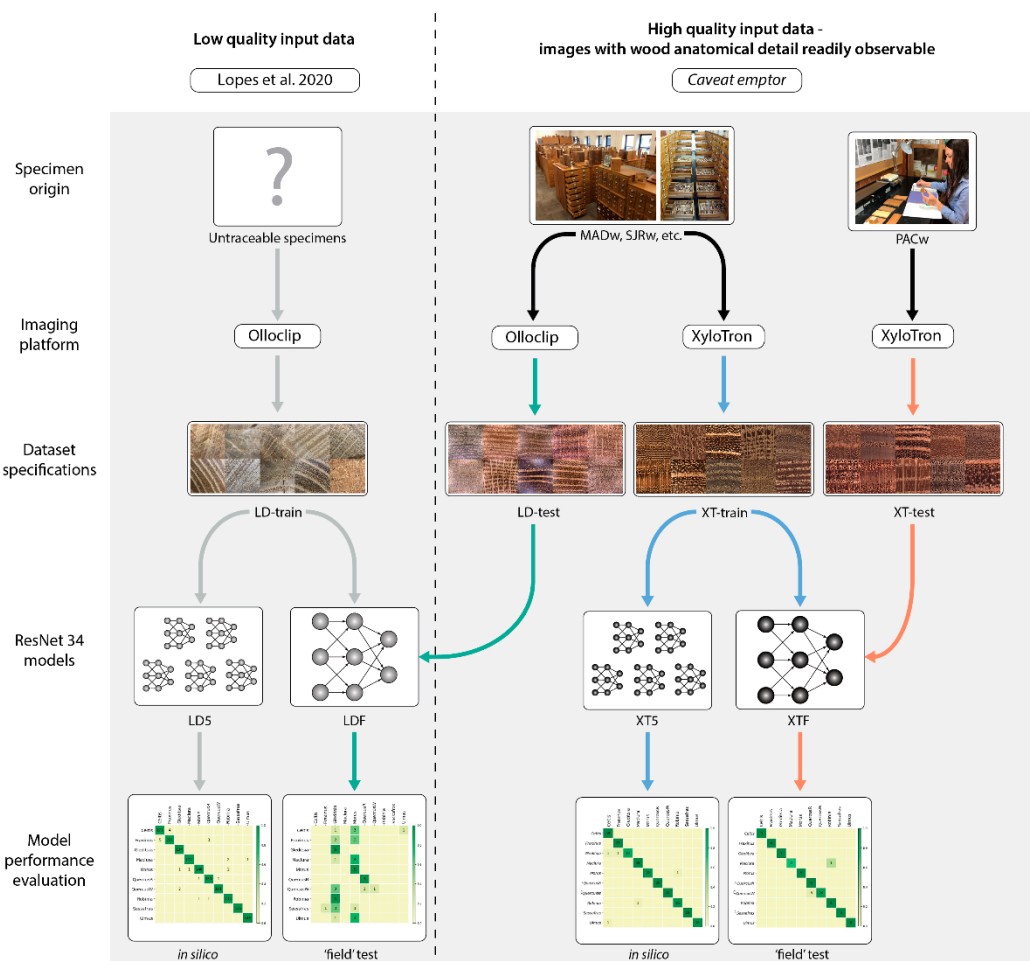

**Figure 3.** A schematic of the project workflow and the relationships between datasets, models, and evaluations.

## 3. Results

### 3.1. Evaluation of LD-Train Image Quality

Table 2 presents summary data for the evaluation of the images comprising LD-train. Robinia and Celtis both contained a minority of images that were not wood anatomically consistent with the class. In Morus, a majority of the images were not anatomically consistent with the class. For all other classes, all images were wood anatomically consistent with the class. Within Morus, 52.0% of the images were consistent with *Catalpa*, a ring-porous wood (readily distinguishable from *Morus*) not included in the model reported in [37], 3.6% of the images were consistent with Maclura, a class that is included in the model. Less than 1% of the images in Morus were inconsistent with the class but otherwise not identifiable. Only in one class, Ulmus, were all the images at least one-quarter knife-cut; in Robinia, Maclura, QuercusR, Fraxinus, and QuercusW a small minority of images showed at least one-quarter of the image with a knife cut. Over three-quarters of the images for Robinia, QuercusR, Fraxinus, and QuercusW showed no relevant wood anatomical detail. Maclura, Morus, QuercusR, Fraxinus, and QuercusW (that is, half of all classes) showed five or more image-pairs—duplicate images that were either horizontally and/or vertically flipped, rotated by 90-degree multiples, or highly overlapping versions of each other.

**Table 2.** The class-wise and average proportion of claims about LD-train images. For the claims "Knife/razor cut", "No anatomy evident", "Wood anatomy evident", and "Image in focus", the proportions are calculated only for images that are consistent with the class. "Prop. of claims" excludes the "No Anatomy Evident" claim. * The entries in the "Duplicate image pairs" row are counts.

| | Celtis | Fraxinus | Gleditsia | Maclura | Morus | Robinia | QuercusR | QuercusW | Sassafras | Ulmus | Average |
|---|---|---|---|---|---|---|---|---|---|---|---|
| Correct taxon | 0.96 | 1 | 1 | 1 | 0.44 | 0.99 | 1 | 1 | 1 | 1 | 0.94 |
| Knife/razor-cut | 0.99 | 0.07 | 0.90 | 0 | 0.80 | 0.02 | 0.18 | 0.05 | 0.94 | 1 | 0.50 |
| No anatomy evident | 0 | 0.88 | 0.08 | 0.04 | 0.39 | 0.81 | 0.77 | 0.83 | 0.06 | 0 | 0.39 |
| Wood anatomy evident | 0.97 | 0.04 | 0.48 | 0 | 0.18 | 0.01 | 0.06 | 0.04 | 0.11 | 0.97 | 0.29 |
| Image in focus | 0.85 | 0.93 | 0.59 | 0.95 | 0.28 | 0.94 | 0.88 | 0.89 | 0.79 | 0.96 | 0.81 |
| Image size 3024 × 3024 | 1 | 0.98 | 0.98 | 0.83 | 0.76 | 1 | 0.08 | 0.94 | 1 | 1 | 0.86 |
| Prop. of all claims | 0.95 | 0.61 | 0.79 | 0.56 | 0.49 | 0.59 | 0.44 | 0.58 | 0.77 | 0.98 | 0.68 |
| Duplicate image pairs * | 0 | 19 | 0 | 16 | 5 | 0 | 13 | 14 | 0 | 0 | |

### 3.2. Evaluation of Models Trained on LD-Train

The image-level cross-validation accuracy of LD5 was 98.5% (prediction confusion matrix in Figure 4, (left)), while the accuracy of the InceptionV4_ResNetV2 model reported in [37] was 92.6%. The lack of specimen level information for LD-train precluded the computation of specimen level metrics. LDF had a specimen level accuracy of 32.0% when tested on LD-test and the corresponding prediction confusion matrix is shown in Figure 4 (right).

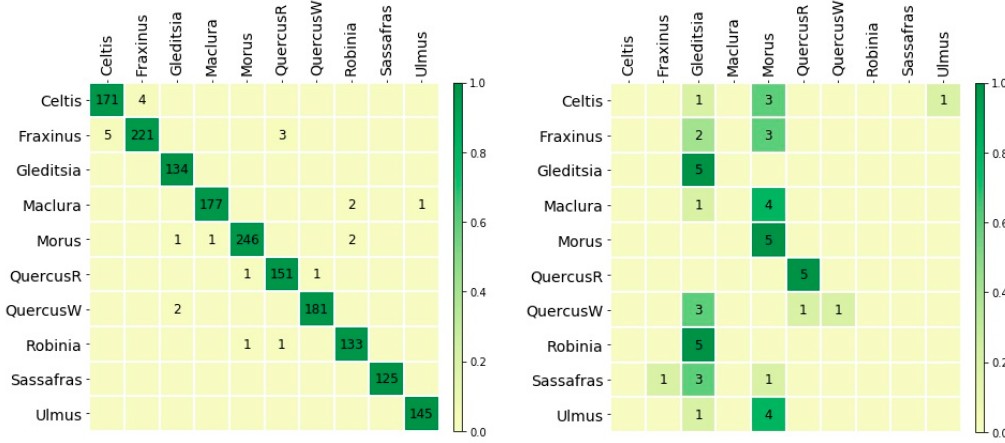

**Figure 4.** Confusion matrices for models trained on LD-train. (**Left**) Image-level confusion matrix for LD5. The confusion matrix, and accuracy (98.5%), were obtained by aggregating over the five cross-validation folds. (**Right**) specimen-level confusion matrix for LDF on LD-test images with an accuracy of 32.0%. The only classes with correct predictions were Gleditsia, Morus, QuercusR, and QuercusW.

### 3.3. Evaluation of Models Trained on XT-Train

The image and specimen level prediction confusion matrices for XT5 and XTF are provided in Figure 5. The predictive accuracy metrics for all the trained models are summarized in Table 3.

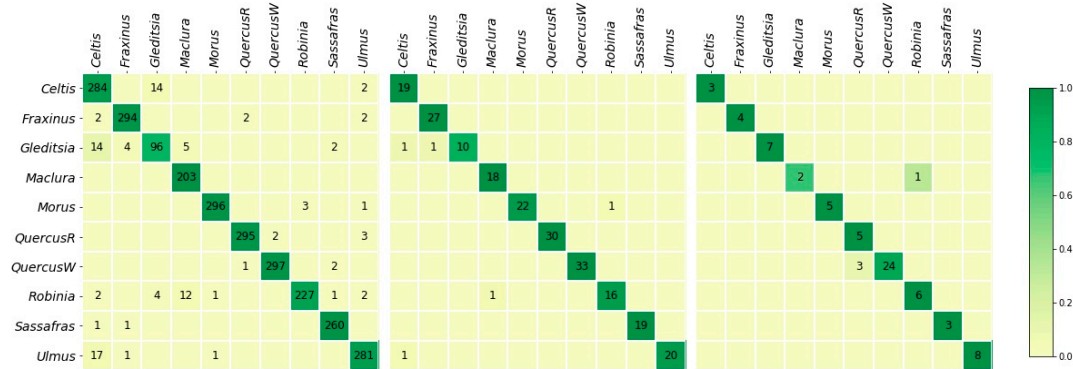

**Figure 5.** Confusion matrices for models trained on XT-train. (**Left**): Image-level confusion matrix for XT5 (accuracy: 96.1%). (**Middle**): Specimen-level confusion matrix for XT5 (the best of 10 random selections of 5 images per specimen, accuracy: 97.1%). (**Right**): Specimen-level confusion matrix for field model XTF predicting on XT-test (accuracy: 94.4%). Cross-validation confusion matrices were obtained by choosing up to five images per specimen and aggregating them over the five folds (also see supplementary materials).

**Table 3.** Image and specimen level model prediction accuracies. The shaded rows report accuracies for proxy field testing and the unshaded rows report the cross-validation accuracies. The cross-validation accuracy reported in [37] was 92.6%. * The average accuracy (standard deviation: 0.5%) over 10 repeats of the process of selecting up to 5 images per specimen. Confusion matrices for each of the 10 repeats are presented in the Supplement.

| Training Dataset | Model | Testing Dataset | Reporting Level | % Accuracy |
|---|---|---|---|---|
| LD-train | LD5 | LD-train | Image | 98.5 |
| LD-train | LDF | LD-test | specimen | 32.0 |
| XT-train | XT5 | XT-train | Image | 96.1 |
| XT-train | XT5 | XT-train | specimen | 97.1 * |
| XT-train | XTF | XT-test | specimen | 94.4 |

## 4. Discussion

### 4.1. High Accuracy Does Not a Valid Model Make

Despite LD5 (accuracy: 98.5%) "outperforming" the [37], InceptionV4_ResNet model (accuracy: 92.6%), the considerable drop in accuracy of LDF (accuracy of 32% when tested on the independent, well-prepared specimens of LD-test) suggests that the model does not generalize well, thereby rendering it essentially non-functional in the real world. We assert that any models developed with LD-train are invalid due to the following fundamental flaws:

1. The foundational requirement for a valid ML methodology is the clear separation between the training and testing sets. For CVWID this requires that every wood specimen must contribute images to exactly one of the cross-validation folds. The presence of sufficient numbers of duplicate images (rotated and cropped versions of images of the same wood tissue) in LD-train violates this foundational requirement *both* at the specimen and image levels, thereby not allowing scientifically valid evaluation such as cross-validation analysis.

2. The lack of adequate specimen surface preparation to expose the relevant wood anatomy (use of saw-cuts instead of knife-prepared) and the presence of incorrectly labeled images (e.g., the majority of the images in class Morus were in fact images of *Catalpa* sp., a taxon not included in the study) are other reasons that make LD-train unsuitable for comparative model performance analyses, and this unsuitability extends to the validity of the initial work as published (Table 2). In contrast, the difference in cross-validation and proxy field testing accuracies for the models trained

on XT-train, XT5, and XTF, is acceptable and the model(s) can be deemed practically functional, especially in human-mediated scenarios.

The lack of clarity regarding specimen level mutual exclusivity in prior works generally, though highlighted by [23] and acknowledged by [49] for the microscope image dataset of [5], has not been fully explored. In addition, the visibility of relevant wood anatomical detail in prior image datasets has also not been investigated. While beyond the scope of this paper to address all available datasets, it would be a worthwhile endeavor for a domain expert to evaluate the quality and validity of any available dataset prior to conducting subsequent analyses.

It should be noted that when proceeding from five-fold cross-validation (XT5) to surrogate field testing in our CVWID model (XTF) for these ten classes, we see a drop of ~3% in the predictive accuracy—this "deployment gap" has been reported in prior works [13,50] but still results in practically useful models for human-in-the-loop settings. For models learned using LD-train, the deployment gap is over 66%, essentially rendering meaningless any claims of deployability in [37].

An important limitation of our datasets is that they were obtained from xylarium specimens and may not capture all the distributional shifts that can manifest in real-world commercial specimens due to natural inter- and intra-class variability, sample preparation and imaging artifacts, and operator skill, and will be an important challenge for the realization of practical field-deployable CVWID. While the XT-train (219 specimens) and XT-test (71 specimens) are the largest data sets used for training and evaluating a CVWID model for the considered classes to date, they may well not represent the entire wood anatomical variation of each class. In the future, augmenting xylarium specimen datasets with images from forensically verified commercial specimens could help address this limitation.

### 4.2. Towards Effective Field Testing and Deployment

The gold standard for CVWID evaluation that establishes its context-dependent sufficiency is field testing, and valid field testing is predicated on forensic validation of field specimens. One cannot know if the model was accurate in the field in the absence of independent, scientifically robust identification (as in [51] which employed human verification of field-collected images] of the field specimens. As noted in [16], model performance evaluation using cross-validation methodology and the surrogate field-testing approach employed only xylarium specimens, which can differ significantly from those found in trade based on factors such as the maturity of the tree, moisture content, growing conditions, and individual tree selection (ref. [52] has a discussion of trained classifier generalization in the context of CV more generally). The impact of these factors on the transition to the real-world deployment of trained models for field testing is largely unexplored.

Evaluation metrics for a real-world deployed model are dependent on whether the goal is field screening or forensic identification, with most compliance/legal enforcement requiring field screening to establish probable cause and subsequently initiate processes for forensic identification to reach a final decision [53]. In this two-step process, the goal would be to achieve context-dependent practical levels of sensitivity and specificity (i.e., determining the deployment context-specific balance between false positives and false negatives) so that costlier downstream forensic identification processes are optimally initiated. Ensembling approaches at the model, patches-per-image, images-per-specimen, and specimens-per-shipment level can improve the robustness of models in the laboratory and in the field, but establishing the correct recipe is likely to be context-dependent and is a critical open problem for the realization of practical CVWID systems (and for any wood identification system in general).

The current state-of-the-art of CVWID suggests that a human-in-the-loop approach, employing field inspectors with a modicum of wood anatomy training, has immediate viability [16]. For effective incorporation of CVWID systems in such scenarios, the design and implementation of intuitive and clean user interfaces that enable operators to access and visualize model predictions, exemplar images, and associated product claims are also critical.

While top-k predictions of a model (and its accuracy) reported in prior works (e.g., [13]) may be useful for such human-in-the-loop settings, as they enable human operators to identify and flag gross errors in the model predictions, their usefulness is dependent on the number, the inter-class anatomical similarity, and the intra-class anatomical variability of the taxa included in the model. Thus, the effective design and enforcement of compliance policies for wood product value chains are dependent on financial, environmental, technological, social, personnel, and political concerns. Given this complex interplay of factors, it is critical to establish a minimal set of necessary baseline quality standards to avoid the pitfalls that can reduce the scientific value of CVWID studies.

We reiterate now that effective data-driven CVWID models must use feature representations that encapsulate the discriminative wood anatomy needed for identification while being robust to the typical/expected variations for the considered taxa. When the dataset used for model development is based on just a few xylarium specimens, with images usually taken within a small area of a specimen surface, it is unclear how much of the expected variability is captured in the dataset (and hence learnable by the model). The question of optimal construction of datasets (i.e., what is the approximate number of specimens required to capture the expected variability within the taxa) for the training and (*in silico*) evaluation of models that generalize well during field evaluation is still open, though [33] indicate at least 10 specimens, and at least 100 images based on a CVWID study of rosewood. Practical approaches for addressing this challenge may include techniques suitable for small-to-medium sized labeled datasets (e.g., transfer learning [44], few-shot learning [54,55], self-supervised learning [56,57]), datasets with carefully chosen context-dependent combinations of xylarium and market specimens, and the pooling of cross-compatible datasets from different CVWID platforms. These goals could be achievable by the community of researchers interested in advancing CVWID by adhering to practices consistent with baseline quality standards.

### 4.3. A Baseline Quality Standards Checklist

Given the logistical challenges involved in performing field evaluation of CVWID, some criteria and practices for developing, evaluating, and iterating scientifically valid, useful in silico models that can have a smooth transition to acceptable field-operational systems include:

1.  the use of multiple unique wood specimens of a sufficient number to capture the wood anatomical variation inherent to the taxa considered, with uniqueness interpreted as being sourced from distinct trees, (see Figure 1 for valid and invalid scenarios),
2.  the consistent, repeatable, and adequate preparation of specimen surface(s) (e.g., sanding, razor cuts) to expose the relevant wood anatomy needed for identification,
3.  the use of an imaging sensor with a sufficient (and reported) spatial resolution to enable the capture of coarse and fine, taxon-dependent, discriminating wood anatomical features,
4.  the acquisition of in-focus images with controlled, repeatable, and consistent illumination,
5.  minimal to no overlap among multiple images from a specimen so that intra-specimen variation is optimally and maximally captured,
6.  the label space design, i.e., the classes (sub-generic, multi-generic, individual species, anatomical characters, character states, etc.) into which the taxa are categorized (model outputs), represent the wood anatomy of the considered taxa while being relevant to the deployment context for the model,
7.  the evaluation of trained models, at the specimen level, on specimens that did not contribute images for model training and/or testing on a completely new set of verified specimens, and,
8.  the parsing of model evaluation results (e.g., confusion matrices) using domain expertise (not all errors are considered equal e.g., out-of-genus identification errors may be worse than congeneric errors [14]), in addition to reporting standard metrics

such as (specimen level) accuracies, precision-recall, F1 score—these latter metrics do not provide class-wise information about which errors were made.

The presumed goal of CVWID systems is the development of models for the classification of specimens into anatomically informed classes of woods that generalize well to new specimens from trees that did not contribute data to the training process, and further that these models be used in the real world, not merely as in silico theoretical case studies. Given this requirement, and unless and until robust data demonstrate otherwise, it is necessary to assume that individual organismal level mutual exclusivity between the training and evaluation datasets (i.e., specimen level mutual exclusivity where each specimen is from a different tree) must be maintained to enable valid in silico model performance that translates to deployable CVWID systems with acceptable real-world performance. Baseline standards 1 and 7 encapsulate this core requirement. Adherence to standard 1 is mandatory and a study that violates it is scientifically invalid. Providing a quantitative threshold for sufficiency in baseline standard 1 is highly challenging, but for valid k-fold cross-validation one requires a minimum of k specimens per class with higher multiples of k likely to provide a more robust model generalization. Good proxy field performance has been demonstrated with moderate-sized datasets (tens of specimens per class) for different sets of woods [12,50]. It is likely that the actual number of specimens and images needed per class will depend on the wood anatomical variability of the class, the relative anatomical similarity between classes in a model, the granularity of the identification task, and the sample preparation and imaging protocols employed. The motivation for standard 2 is the requirement to capture relevant wood anatomical details which can be learned by the model and generalized to specimens in the field and is not a human perceptual quality requirement. It does not preclude the use of razor/knife cuts for surface preparation [58], the use of appropriate image compression methods, or the use of representations that sacrifice perceptual quality for higher model performance [59] so long as the necessary wood anatomical features can be learned. Datasets that neglect standards 2, 3, and 4 can lead to trained models that have learned non-relevant features that do not generalize well, and following them provides assurance that wood anatomical features are visible and that datasets have the potential to be combined or compared if images of similar spatial resolution are made available in separate studies. Standard 5 can be considered an image-level version of the mutual exclusivity called for in standard 1, and it reduces the risk of learning features from the same image patch, which may lead to models that do not capture the variations in the anatomical features that are typical for the taxa considered. Standard 6 increases the likelihood that a given model addresses practically useful requirements—the scientific and practical value of models with ill-posed CVWID classes is questionable at best. Standard 7 ensures that a model has learned generalizable, useful features rather than only details of the dataset. Standard 8 ensures that the kinds of errors made by a model are considered when evaluating efficacy, as some kinds of errors might be more detrimental than others, for example, confusing an endangered wood for a non-endangered one.

### 4.4. Revisiting Context-Dependency in CVWID and Other Modalities

Although we have placed our primary emphasis on data quality, data hygiene, model development, and model evaluation thus far, the context-dependency of CVWID (standard 6, above) is foundational to developing useful models. Just as flawed data can only yield flawed or fictional results, ill-considered class labels and oversimplified label space granularity will necessarily lead to results of little or no practical relevance or utility. For example, [60] use a single species, *Cedrela odorata*, as a proxy for all woods controlled by the Convention on the International Trade in Endangered Species of Wild Fauna and Flora (CITES), then use five anatomically disparate woods to comprise a single contrast non-CITES class. They then test this binary model on a set of eight additional non-CITES woods not included in either class (Figure 6, left), and, inexplicably, they do not include any additional CITES woods in their subsequent model evaluation. Figure 6 (right) clearly shows that various CITES-controlled woods are wood anatomically disparate from *Cedrela*

and from each other, and some are anatomically similar to the outgroup taxa used to comprise the contrast class (Figure 6, left)—an approach that is botanically, forensically, and logically flawed by virtue of being contextually uninformed.

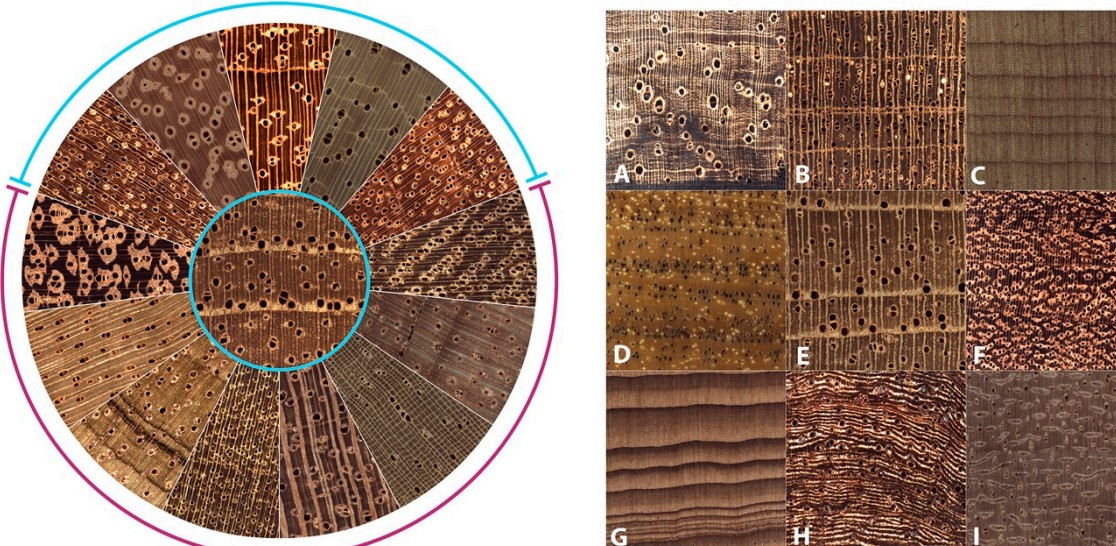

**Figure 6.** XyloTron images of the transverse surfaces of woods relevant to [58]. Images on the right represent 6.35 mm of tissue on a side; images on the left were captured at the same magnification and resolution. (**Left**): The woods used in [58]. The blue-bordered center image is *Cedrela odorata*, the single-species CITES-proxy class. The five woods bordered by the blue arc together comprise the non-CITES class. The red-bordered woods are the eight other species used to "test" the binary CITES-proxy vs. non-CITES model. Note that woods bordered by the blue arc are not anatomically similar to *Cedrela odorata*, but some show anatomical similarity with the woods comprising the non-CITES class. Right: *Cedrela odorata* (**E**) and eight other CITES-listed woods (**A–D**, **F–I**). Note that the wood anatomical variability of this small subset of CITES-listed woods is greater than that of the non-CITES woods used in [58]. (**A**), *Dalbergia nigra*. (**B**), *Swietenia macrophylla*. (**C**), *Araucaria angustifolia*. (**D**), *Guaiacum sanctum*. (**F**), *Pericopsis elata*. (**G**), *Abies guatemalensis*. (**H**), *Pterocarpus santalinus*. (**I**), *Aquilaria malaccensis*.

While the promise of CV/ML approaches for wood identification has been demonstrated many times in the past two decades, the development and analysis of wood anatomy-informed, practically relevant models are generally lacking. The baseline quality standards above are a starting point for the design, implementation, and evaluation of CVWID models so that their in-laboratory performance generalizes sufficiently to a real-world, field setting. Models trained, evaluated, and compared on datasets that violate the above requirements for scientific validity, can only be deemed inherently invalid, despite high reported in silico accuracy. We assert that adherence to the minimal set of baseline quality standards espoused above will help avoid the perils and pitfalls of garbage-in, fiction out (GIFO) hypothesized in [18] and demonstrated here, and will allow "fair" comparisons of competing models and/or technologies generated from scientifically valid datasets.

It should be noted that the standards proposed have applicability beyond CVWID and can be used to evaluate results reported using other modalities. For example, the recent work of [61] reports an almost "perfect" classification of more than 48 wood species using X-ray fluorescence (31 species had exactly one specimen that contributed data to both the training and testing data, and the remaining 17 species were represented by 2 specimens each, and specimen mutual exclusivity was not imposed)—a result that is highly questionable given the lack of specimen level separation between training and testing data. Our recommended quality standards are an explicit enumeration and a

reiteration of "obvious", but oft-violated, ML principles, but they are applicable to the evaluation of wood identification systems in a modality-agnostic fashion.

## 5. Conclusions: *Caveat emptor*

Selecting among competing CVWID systems has the potential to be a difficult burden for policy or enforcement/compliance officers who are not themselves domain experts in wood identification and ML, and it would be natural to assume that reported model accuracy is a valid metric for comparison. The high in silico accuracies reported by virtually all CVWID models suggest that a rigorous, baseline framework is required for interpreting published performance metrics guiding future progress towards field-operational CVWID systems. To improve a broader understanding of CVWID, the foundational ML and wood anatomy aspects of CVWID were highlighted as a collection of baseline quality standards and the need for such was illustrated by the novel presentation of two detailed case studies: a critical revisitation of prior work by [37], and a new complementary study of the same woods that exemplifies the baseline quality standards that were espoused. By adhering to the baseline quality standards outlined here, future CVWID studies should be able to avoid the dangers of GIFO and thereby achieve greater real-world relevance, whereas failure to do so is likely to result in spurious in silico results that do not generalize well to the real world. Only by testing models on unique specimens (e.g., real-world testing) can generalizability be evaluated. It seems increasingly clear that the successful implementation of CVWID systems depends on the collaboration of machine learners, wood anatomists, policy-makers, and enforcement officials and we believe that the minimal set of proposed baseline quality standards, based on the maxim "Models without (good) data are fantasy (adapted and modified from [62])", can be used by the stakeholders to evaluate prior and future works in CVWID and similar technologies (e.g., [59,63]).

**Supplementary Materials:** The following supporting information can be downloaded at https://www.mdpi.com/article/10.3390/f13040632/s1: label space design details (Table S1), class wise central patch montages (Figures S2–S21), imaging parameters (Figure S1, Table S2), and additional confusion matrices (Figure S23).

**Author Contributions:** Conceptualization, P.R. and A.C.W.; methodology, P.R. and A.C.W.; software, P.R.; validation, P.R. and A.C.W.; formal analysis, P.R. and A.C.W.; investigation, P.R. and A.C.W.; resources, A.C.W.; data curation, A.C.W.; writing—original draft preparation, P.R. and A.C.W.; writing—review and editing, P.R. and A.C.W.; supervision, A.C.W.; project administration, A.C.W.; funding acquisition, P.R. and A.C.W. All authors have read and agreed to the published version of the manuscript.

**Funding:** This work was supported in part by a grant from the US Department of State via Interagency Agreement number 19318814Y0010 to ACW and in part by research funding from the Forest Stewardship Council to ACW. PR was partially supported by a Wisconsin Idea Baldwin Grant.

**Institutional Review Board Statement:** Not applicable.

**Informed Consent Statement:** Not applicable.

**Data Availability Statement:** The datasets presented in this article are not immediately available, but a minimal data set can be obtained by contacting the corresponding author; the full data set used in the study is protected for up to 5 years by a CRADA between FPL, UW-Madison, and FSC. Requests to access the datasets should be directed to corresponding author.

**Acknowledgments:** We are grateful to Sarah Friedrich of the UW-Madison Department of Botany for her patience (with us) and skill in preparing Figures 1, 3 and 6 and to Lauren Vance for being willing to put her other project on hold and collect the LD-test dataset using her phone. This manuscript benefited greatly from the comments of four reviewers, Phil Townsend, Samuel Glass, Diego Patiño, Ganesh Sankaralingam, and from suggestions from Thomas Eberhardt.

**Conflicts of Interest:** The authors declare no conflict of interest.

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
