# Peer review of "Caveat emptor: On the Need for Baseline Quality Standards in Computer Vision Wood Identification"

_forests, doi:10.3390/f13040632_

Round 1

Reviewer 1 Report

Overall, the journal article is very well written and informative.

However, the authors seem to use high-falutin words that are excruciatingly unnecessary. This muddles the meaning of each sentence. Please revise entire paper by using common jargon language to reach a diverse audience, otherwise you are scrutinizing and alienating people who are less familiar with this kind of research and words.

The overall length of the journal article and the supplemental material is excessively long (>47 pages in total). I recommend to the authors that they drastically shorten the article. In addition, most of the supplemental material does not really bring anything more to the paper. I recommend to the authors consider reducing the supplemental material too.

There are many places where the authors need to cite literature especially when they mention "prior works". If these "prior works" statements are not cited, it is considered plagiarism.

Due to a very short and limited time frame for review of the 20 pages of this article and the additional 27 pages of supplemental material all aspects of these documents were reviewed.

Nevertheless, if the authors take all the helpful edits and notes in this document and apply it to the supplemental pages, it should help improve it for future publication. In its current state, there are too many in consistencies and errors. 

A major flaw in this study was that when the authors repeated the study from Lopes et al. (2020), they did not use the same methodology, nor same number of images/sizes. To properly conduct research or repeat research, one would needs to be very similar to the previous study by using the same methodology, same sample size, etc. otherwise, it is comparing apples and grapes.

The authors also mentioned for reviewers not to muddy or dilute the quality and reputation of their journal article by recommending their own journal article(s), however, the authors self-cited themselves repeatedly. A question for the authors to consider is how come the authors muddy and dilute the resource quality and reputation of this journal and their own journal article by self-citing themselves over and over? This is not very professional, especially since there are other literature specifically on this subject matter for the authors to cite.

Author Response

Please find our replies in the attached document. Thank you for the detailed review.

Reviewer 2 Report

The proposed work is interesting and significant to the domain of study.

All section has been discussed and explained well, especially the results section. If possible, the author should add a more detailed explanation on model architecture and training section - maybe can include one figure on CNN architecture that they used for this research. The author should also highlight why the model has been chosen to train both datasets. 

Please check the formatting - the test on page 10 is not well-formatted (under figure 4)

For the discussion section, the authors have discussed and justified the results very well; just a suggestion:  if possible, authors should consider dividing the discussion according to subtopics.  

Author Response

Thank you providing the review. Please find our replies in the attached document.

Reviewer 3 Report

Interesting review and appreciate the experiment done to prove the suggested solution.

The article is very well written and already reviewed by many experts as stated in the manuscript. As the nature of the article is not a novel algorithm but merely a suggestion and example on how to systematically run wood identification experiments, I find myself agreeing on the author's points. He managed to state the obvious in a clear writing style and in a systematic way.

Line 273: Do the authors work with a wood expert to confirm the species and good quality of all four databases used in the experiment? Maybe some details on this wood expert can be included in the article.

Author Response

Thank you for your review. Please see the attached document.

Round 2

Reviewer 1 Report

Despite an effort on the authors’ corrected article, based on reviewers edit suggestions, they did not show improvement on their journal article. It is ok that the authors do not accept every suggestion made by reviewers; however, these authors barely used 15-20% of the edits and comment suggestions made. Therefore, little improvement was done on this article, not warranting publication at this time. The authors have a good article, but it has lots of flaws and areas where it needs to be improved upon. The authors need to take time and carefully review, for a second or third time, the reviewer’s document that was attached during the first peer-review. Major concerns that have still not been addressed are that the authors self-cited themselves repeatedly. As the authors muddy and dilute the resource quality and reputation of this Forest journal and their own journal article by inappropriately self-citing themselves repeatedly. This is not professional, especially since there are other literatures specifically on this subject matter for the authors to cite. Another major concern is plagiarism. The authors use these two phrases - “see text for exceptions” or “prior works”, but they do not give specific or actual references. No one knows which reference the authors are referring to as this is just eluding to random published articles. A professional journal article needs to be as clear as possible for the readers and audiences. In these cases, the reader does not know which specific “prior works” the authors are referring. These areas need citations; otherwise, it is plagiarism for taking others ideas without citing them. I do hope the authors greatly improve these and other major areas of concern to have an improved professional journal article in the future. Best regards.

Author Response

In this re-review, the reviewer has not identified any scientific, methodological, or technical weaknesses in our work. We think that you will see in our reply to their first review, we addressed all of the substantial edits/comments made, either by altering our manuscript or explaining why doing so would be incorrect or would conflict with our data and/or norms of scientific publishing.  Author Wiedenhoeft has been an Associate Editor for one or more international journals since 2003, and does not find Reviewer 1’s comments about “plagiarism” as having scientific or editorial merit – could you please review their comments, our responses, and our manuscript and provide guidance?

With regard to self-citation, our lab is the most active group publishing this kind of work on CVWID using macroscopic images and pretending otherwise would be disingenuous.  Despite this, the great minority of papers referenced in our manuscript (less than 15%) are self-citations, so it is difficult to understand the reviewer’s complaint here.